# Therapeutic Approaches in Pulmonary Arterial Hypertension with Beneficial Effects on Right Ventricular Function—Preclinical Studies

**DOI:** 10.3390/ijms242115539

**Published:** 2023-10-24

**Authors:** André Balsa, Rui Adão, Carmen Brás-Silva

**Affiliations:** 1Cardiovascular R&D Centre—UnIC@RISE, Department of Surgery and Physiology, Faculty of Medicine, University of Porto, 4200-319 Porto, Portugal; up201705745@up.pt (A.B.); ruiadao@med.up.pt (R.A.); 2Department of Pharmacology and Toxicology, School of Medicine, Complutense University of Madrid, 28040 Madrid, Spain; 3CIBER of Respiratory Diseases (CIBERES), 28029 Madrid, Spain; 4Faculty of Nutrition and Food Sciences, University of Porto, 4150-180 Porto, Portugal

**Keywords:** pulmonary arterial hypertension, pulmonary arterial banding, right ventricle, preclinical studies

## Abstract

Pulmonary hypertension (PH) is a progressive condition that affects the pulmonary vessels, but its main prognostic factor is the right ventricle (RV) function. Many mice/rat models are used for research in PAH, but results fail to translate to clinical trials. This study reviews studies that test interventions on pulmonary artery banding (PAB), a model of isolated RV disfunction, and PH models. Multiple tested drugs both improved pulmonary vascular hemodynamics in PH models and improved RV structure and function in PAB animals. PH models and PAB animals frequently exhibited similar results (73.1% concordance). Macitentan, sildenafil, and tadalafil improved most tested pathophysiological parameters in PH models, but almost none in PAB animals. Results are frequently not consistent with other studies, possibly due to the methodology, which greatly varied. Some research groups start treating the animals immediately, and others wait up to 4 weeks from model induction. Treatment duration and choice of anaesthetic are other important differences. This review shows that many drugs currently under research for PAH have a cardioprotective effect on animals that may translate to humans. However, a uniformization of methods may increase comparability between studies and, thus, improve translation to clinical trials.

## 1. Introduction

Pulmonary hypertension (PH) is a progressive condition that affects pulmonary vessels, leading to worsening of the right ventricular (RV) function [1], which is the main prognostic factor [2]. It is defined as an elevation of mean pulmonary arterial pressure above 20 mmHg at rest [3]. There are five major types of PH, based on clinical presentation, pathophysiology, and management [3]. Group 1 PH is also called pulmonary arterial hypertension (PAH) and it is less common than PH groups 2 and 3 [3]. Most (50–60%) PAH cases are idiopathic [3]. The other most common associated conditions are connective tissue disease (such as systemic sclerosis), congenital heart disease, and portal hypertension [3,4]. Treatment of underlying conditions is possible for patients with PH group 2 (PH associated with left heart disease), group 3 (PH associated with lung diseases/hypoxia), and group 4 (PH associated with pulmonary obstructions) [3], unlike for most patients with PAH. Current treatment for PAH consists of general measures, calcium channel blockers (in responders—less than 10% of affected people), and PAH therapies (endothelin receptor antagonists—ERA, phosphodiesterase 5 inhibitors—PDE5i, and prostacyclin agonists) [3,4]. If response is inadequate, patients can undergo interventional therapies, such as lung transplantation [3,4].

Many small animal models are used for research in the field of PH: from older, “classical” models—chronic hypoxia (CH) and monocrotaline (MCT)—to newer models, such as Sugen5416/hypoxia (SuHx) and pulmonary artery banding (PAB) [5]. “Classical” models induce a milder phenotype [5]. The MCT model is induced by a single injection of MCT, which leads to PH, RV hypertrophy, and pulmonary vascular remodelling—as in human PH—but it also affects the liver, the myocardium, and the kidney, unlike in human diseases [6]. CH animals are exposed to a hypoxic (generally with 10% oxygen) environment for 3–4 weeks [6]. This causes pulmonary vascular remodelling, which improves with normoxia [6]. Therefore, they are mostly used to investigate milder forms of PH, like group 3 PH [7]. SuHx animals receive an injection of a vascular endothelial growth factor receptor 2 (VEGFR-2) antagonist (semaxinib or Sugen5416) and then are exposed to hypoxia, like CH animals [6]. SuHx rats have the advantage of showing pulmonary plexiform lesions—like human PAH—as well as vascular remodelling [6]. PAB rats or mice undergo surgery to permanently constrict the pulmonary trunk, which leads to RV remodelling without PH [7]. They are used to evaluate the direct effects of drugs on the RV [7].

Many drugs which improve PAH in small animals fail in clinical trials [8]. In fact, in a recent meta-analysis, only 41 out of 522 interventions in animal models (8%) were ineffective [9]. Recently, more and more progress has been made, as PAH is linked to immune system dysregulation, microRNA profile changes, and other types of gene expression regulation [10]. However, only drugs targeting three pathways are currently approved for PAH treatment—nitric oxide, endothelin, and prostacyclin pathways—and they are all related to benefits in pulmonary vasculature [8]; no approved therapy primarily targets the RV [11]. This difficult translation from animal models to humans has multiple explanations. Most importantly, no existing animal model replicates all features of PAH in humans [6]. Some problems are a milder phenotype (CH), damage in other organs (MCT), and an absence of pulmonary vessel remodelling (PAB) [7]. Also, depending on the model and on the methodology—type of rat/mouse, induction duration, or anaesthetic used for hemodynamic evaluation—the phenotype can vary greatly [5,11].

The current PH animal models have similarities and differences to human PAH. Therefore, it is advantageous to use models which combine more than one hit (like SuHx), or to compare the effect of pre-clinical drugs on multiple models [12]. Furthermore, as the main prognostic factor of PAH is the RV function [2], direct cardioprotection—assessed by PAB—is an interesting novel option [7], and recent studies have concomitantly evaluated potential PAH drugs in PAB plus at least one PH model. This review summarizes and analyses these studies. We aim to show the effects of multiple drugs on different animal models, with a special focus on PAB.

## 2. Results

Nineteen studies were selected after applying the exclusion and inclusion criteria (Figure 1) [13,14,15,16,17,18,19,20,21,22,23,24,25,26,27,28,29,30,31]. They were published between 2009 and 2022, providing data for twenty drugs and two combinations of two drugs, with a great variety of mechanisms of action (Table 1). Some drugs are being clinically used for PAH—such as sildenafil, tadalafil, and macitentan—and others are important in other diseases—such as sacubitril/valsartan, dapagliflozin, and ivabradine.

### 2.1. Methodology

Drugs were tested using five different PH/RV remodelling models: chronic hypoxia (CH), SU5416/hypoxia (SuHx), monocrotaline (MCT), monocrotaline + shunt (MCT + S), and pulmonary artery banding (PAB).

All MCT animals were rats (Table 2) and a single MCT dose of 60 mg/Kg was administered, except in one study (30 mg/Kg) [16], to induce PH. In two studies, rats further underwent aortocaval shunt surgery (MCT + S) [19,31]. Other models underwent hypoxia periods: CH and SuHx (Table 3). Almost all these animals were exposed to air with 10% O2 for 3/4 weeks. At the start of the hypoxia period, SuHx rats were additionally injected with a VEGFR inhibitor—Sugen 5416 (semaxinib)—at a 20 or 25 mg/Kg dose. PAB animals were submitted to a surgery for pulmonary artery constriction, whose grade of constriction is defined by the needle/clip size. Even for the same strains of mice/rats, the sizes greatly varied; for example, in Sprague–Dawley banded rats, the needle size ranged from 22 G to 16 G, and one study used a 0.9 mm diameter clip [18].

Treatment regimens also had important variances (Table 2 and Table 3). Some studies adopted a preventive strategy, starting the treatment immediately after induction, while others waited some (maximum 4) weeks to start it—a therapeutic strategy. Treatment duration ranged from 1 to 7 weeks, and treatment duration was the same for PH and PAB models in about half the studies.

Finally, an important variable for the final outcomes is the choice of the anaesthetic for hemodynamic measurements. In this regard, most (at least 10 out of 19) of the experiments used isoflurane (Table 1).

### 2.2. RV Structure

Fulton index and RV fibrosis were the most assessed RV structure parameters (Table 4). Other parameters include cross-section cardiomyocyte area/diameter (CSA/CSD) and serum brain natriuretic peptide (BNP). Most drugs showed beneficial effects in PH models, and nearly half improved RV structure parameters in both PH and PAB models. 

Sildenafil, macitentan, tadalafil, and a combination of the previous two had no effect on the PAB animals except for macitentan, which ameliorated the Fulton index. Sildenafil even significantly increased the RV cardiomyocytes cross-sectional areas of the PAB group, despite decreasing it in the MCT model.

Other drugs had positive effects in both the PH models and PAB, and some improved no parameters (Table 4). Urocortin-2, ivabradine, sunitinib, sorafenib, neuregulin-1, and dantrolene ameliorated multiple parameters in both types of models. Sacubitril/valsartan, dapagliflozin, and RVX208 had no significant effect on these parameters except for RVX208, which further increased Fulton index in PAB rats [31].

### 2.3. RV Systolic Function and Blood Pressure

RVSP and TAPSE were the most assessed RV systolic function parameters (Table 5). Other parameters include the RV ejection fraction (RVEF), cardiac output (CO), cardiac index (CI), and mean arterial pressure (MAP). Most drugs ameliorated RVSP in the PH models. However, out of eighteen drugs/combination of drugs, only one drug (dichloroacetate) improved RVSP in PAB animals [26]. Dichloroacetate also attenuated the pulmonary artery gradient in banded rats [26].

Macitentan, tadalafil, and their combination improved three RV systolic function parameters—RVSP, TAPSE, and CO—in SuHx animals but not in PAB animals, except for macitentan, which also ameliorated TAPSE [22]. Similarly, sildenafil decreased RVSP in a PH model, but not in PAB rats [28].

Ivabradine and MitoQ increased TAPSE in PAB animals, but had no effect in MCT and CH animals, respectively [17,25]. Gallein improved CO only in the MCT group and CI only in the PAB group [27]. Sacubitril/valsartan decreased mean arterial pressure [14] and sorafenib increased it in MCT rats [18]. Of all the drugs with data for both parameters (16 drugs), only dapagliflozin had no significant effect on either RVSP or TAPSE [21].

### 2.4. RV Diastolic Function and Pulmonary Vascular Remodelling

In most (14 out of 19) studies, researchers measured RVEDD and/or RVEDP (Table 6). In two studies, PH induction with MCT injection did not increase RVEDD and RVEDP [16,17]. PAB surgery did not increase RVEDP in one study [13]. Four studies also measured RV tau.

Urocortin-2, sildenafil, and tadalafil ameliorated one or more parameters in the PH models, but not in the PAB animals. Differently, ivabradine and dantrolene reduced RVEDP in the PAB animals, but not in the MCT rats. GapmeR H19 improved RVEDD and RVEDP in both the MCT and PAB animals [24].

Most studies also provided data on pulmonary vascular remodelling (including intimal/wall thickness) and/or hemodynamics (including pulmonary artery acceleration time—PAAT, pulmonary vascular resistance—PVR and total pulmonary resistance—TPR) (Table 7). Data from total, medial, and intimal thickness resulted from the analysis of many different arteriole size ranges. Most (12 out of 14) drugs improved PAAT, PVR, or TPR.

Macitentan and tadalafil, alone or combined, improved TPR—but only when combined significantly decreased arteriole muscularization [22]. Dapagliflozin and GapmeR H19 had no effect on pulmonary hemodynamics parameters and these drugs also did not decrease mean medial thickness.

### 2.5. Summary of Differences across Models

Overall, 63.0% of PH models results are in line with the PAB results (40.3% both improve the same parameter and 22.7% both have no significant effect). This percentage is bigger for structure parameters (results of Table 3, plus RV weight measures, TIMP-1, and RV wall thickness) of the MCT rats (75.8%) (Appendix A). The already approved therapies for PAH—macitentan, tadalafil, and sildenafil—account for nearly half of the discordances (43.9%). Considering all other drugs, overall concordance rises to 73.1%, and for structure parameters in MCT rats it rises to 92.6%.

The most used models, in addition to PAB, were MCT and SuHx. Overall concordance of MCT (65.9%) and SuHx (61.9%) with PAB results was similar.

## 3. Discussion

We found that multiple drugs both improved pulmonary vascular hemodynamics in PH models and ameliorated RV structure and function after PAB in rats and mice. With drugs other than ERA and PDE5i, PH models and PAB frequently exhibited similar results (73.1% concordance), particularly in the case of MCT rats for structure-related parameters (92.6%).

RVSP accounted for most of the differences between PH models and PAB. Only dichloroacetate improved it in banded animals, whereas 14 out of 19 drugs/combination of drugs improved RVSP in the PH models. Results on RV fibrosis, on the other hand, all agreed (12 drugs). ERA and PDE5i—macitentan, sildenafil, and tadalafil—improved most parameters in PH models, but almost none in PAB animals; only macitentan ameliorated two—the Fulton index and TAPSE. A combination of macitentan and tadalafil improved pulmonary remodelling (arteriole muscularization), unlike both drugs in monotherapy. Unexpectedly, dapagliflozin was the only drug that improved no parameters.

Methods between papers greatly varied. The period from induction until the start of the intervention ranged from 0 to 4 weeks in most studies, and, in one study, treatment was started 1 week before PAB surgery. Intervention periods ranged from 2 to 7 weeks. Other examples include the needle size for PAB surgery and the anaesthetic used for hemodynamical evaluation. Additionally, almost all studies used male animals and a MCT dose of 60 mg/Kg.

### 3.1. Recent Studies Show Multiple Drugs with Cardioprotective Potential

PAB is a model used in PH research. Rats or mice undergo surgery to mechanically constrict the pulmonary trunk, developing RV dysfunction, without affecting the pulmonary vessels [32]. Therefore, if a drug improves the RV function of a PAB animal, that drug likely has a direct cardioprotective action [7]. In PH models, an improvement in RV function can also be the result of indirect action, through afterload reduction due to pulmonary effects; therefore, the PAB model is useful to distinguish these effects [7].

As RV function is the main determinant of prognosis in PH [2], cardioprotection is seen as key to improving PH treatment [7]. Therefore, researchers search more and more for drugs with direct benefits on the RV. As the studies in this review are recent (more than half were published after 2018) and they use the PAB model, it is comprehensible that most drugs seem to directly protect the RV. Some documented cardioprotective mechanisms include pathways related to fibrosis (GS-444217, sorafenib, sunitinib), mitochondrial dysfunction (dichloroacetate), oxidative stress (MitoQ), and epigenetic alterations (GapmeR H19, RVW208, sodium valproate) [33].

### 3.2. RVSP Accounted for Most Discordances, RV Fibrosis for Most Concordances

In the absence of RV outflow obstruction, RVSP estimates pulmonary artery systolic pressure, which can be used to calculate mean pulmonary artery pressure (mPAP) [34]. PH models, like in PH in humans, present an elevated mPAP [5]. This explains why, in all studies included in this review, PH models also exhibit an increased RVSP. Treatment with most drugs decreased RVSP, so these drugs also ameliorated mPAP.

In the PAB model, there is an obstruction to RV outflow, leading to RV pressure overload [7], which causes RVSP elevation. All drugs but one had no effect on RVSP. Only dichloroacetate decreased RVSP in the PAB model [26]. Dichloroacetate also reduced the pulmonary artery pressure gradient [26]. This suggests that RVSP decrease was caused by a reduction in the pressure gradient. As PAB causes a fixed constriction on the pulmonary artery, the pressure gradient should also be constant. This finding requires further research.

PH and PAB models exhibited similar RV fibrosis improvements, which suggests that these models may share common fibrotic pathways. Furthermore, 92.6% of MCT results agreed with PAB for the structure parameters, excluding ERA and PDE5i. This disagrees with a recent review on RV fibrosis due to PH, which points to some differences in fibrosis location and mechanisms, although its current understanding is incomplete [35].

### 3.3. Results from Individual Studies Should Be Interpreted with Caution

Other studies tested these drugs on the same animal models and results lacked consistency. In the Schafer et al. study [28], sildenafil treatment for 3 weeks, immediately after PAB surgery in rats, had no effect on the RV function and structure, except for an increase in the cardiomyocyte cross-sectional area. Rai et al. [36] found similar results after 4 weeks of treatment. Borgdorff et al. [37,38] obtained different results, depending on the treatment regimen. Preventive strategy, with 4 weeks of sildenafil from PAB surgery day 1, resulted in RV systolic function improvement, no effect on RV diastolic function, and RV fibrosis worsening [37]. A therapeutical strategy based on sildenafil treatment starting 4 weeks after surgery, for 4 weeks, resulted in RV systolic, diastolic improvement, and RV fibrosis reduction [38]. Studies on MCT rats showed improvements in RV systolic and diastolic function and structure [39,40,41].

Sildenafil, like tadalafil, is a PDE5 inhibitor. PDE5 is an abundant enzyme in the lung vasculature that degrades cGMP [42]. Its inhibition leads to vasodilation, improving pulmonary hemodynamics [42]. There is also evidence of direct cardioprotective effects [43]. Similarly, prostacyclin analogues increase the amount of cellular cAMP, which also has vasodilator properties [44]. A recent study suggested that treprostinil, a prostacyclin analogue, may improve RV disfunction in severe PAH [44]. However, these effects may be of a lesser importance in PAH, as studies with PH models show benefits, but many with PAB reveal an absence of improvements or even worsening of RV fibrosis [22,28].

In other papers, contrary to the Li et al. study [21], dapagliflozin improved RV function, RV hypertrophy, and pulmonary vascular remodeling in MCT rats [45,46]. Tang et al. attributed the difference to the lower mortality of MCT rats and to the longer duration of treatment: 3 instead of 2 weeks [45]. Wu et al. treated rats for even longer: 5 weeks from MCT injection [46].

Andersen et al. reported that sacubitril/valsartan improved some parameters only in SuHx rats and mean arterial pressure also in PAB [14]. In other studies, sacubitril/valsartan also improved the Fulton index, RV wall thickness, fibrosis, and RVEDP in MCT and SuHx animals [47,48]. A study found a RVSP and RV hypertrophy reduction in PAB animals [49].

Like the study included in this review [16]—which tested sodium valproate in MCT and PAB rats—other studies showed beneficial effects of valproic acid on CH and MCT plus CH animals [50,51]. A study reported on the multiple detrimental effects of trichostatin A [52], another histone deacetylase inhibitor, in PAB rats: it worsened fibrosis, RV dilation, CO, TAPSE, and more parameters.

These different findings can be related to the methodology, particularly in the induction of the models. The most important factor for hemodynamic and structural and vascular worsening is the induction period; the longer it is, the more severe the phenotype [5]. Additionally, older models—CH and MCT—cause milder phenotypes, some anaesthetics influence RVSP and mPAP values (greater pressure values are obtained with isoflurane), and preventive strategies lead to better outcomes than therapeutic ones [5]. In the PAB model, a tighter constriction of the pulmonary artery causes a more severe phenotype, ranging from RV adaptative dysfunction to RV failure [53].

### 3.4. Some Drugs Are Already Approved and Other Are Being Evaluated in Clinical Trials

Some of the drugs considered in this review are already approved for PAH. The 2022 ESC/ERS guidelines for PH [3] recommend the use of PDE5 inhibitors and/or ERA in some patients, depending on cardiopulmonary comorbidities and mortality risk, due to many favourable effects in clinical trials. PDE5 inhibitors improve hemodynamics, functional class, and 6 min walk distance [54]. They also reduce mortality [54]. In the REPAIR clinical trial, macitentan (ERA) improved pulmonary hemodynamics, RV function, and structure [55], like in SuHx rats (20). The AMBITION clinical trial compared combination and monotherapy of ambrisentan—another ERA—and tadalafil [56]. Combination therapy further reduced morbidity and improved 6 min distance [56].

Other drugs have already been tested in smaller clinical trials and observational studies, with positive outcomes. Dichloroacetate improved mPAP and PVR in genetically susceptible patients [57]. Sacubitril/valsartan also reduced mPAP in patients with heart failure with reduced ejection fraction (HFrEF) [58] and preserved ejection fraction (HRpEF) [59]. A positive effect on RV function, assessed by TAPSE, was only present in HFrEF [58]. Ivabradine led to functional improvements in 10 PAH patients with high heart rates [60]. Sorafenib showed improvements in patients with refractory PAH [61]. However, in another study without a placebo group, sorafenib led to a decrease in the cardiac index and a non-significant increase in systemic blood pressure [62].

### 3.5. Small Findings Can Be Important

In one article [27], cardiac output (in mL/min) and cardiac index (mL/min/g) were assessed. Gallein treatment improved cardiac output only in MCT rats, and cardiac index only in PAB animals: the bodyweight indexing affected the results. Other studies show significant differences between the control and MCT groups (neuregulin), MCT and MCT + treatment (neuregulin), sham and PAB (ivabradine), and PAB and PAB + treatment (ivabradine). Borderline cardiac output improvements can become significant with or without indexing.

Also, unlike their monotherapy, macitentan plus tadalafil improved pulmonary vascular remodeling in MCT rats [22]. This was the only advantage of the combination. Accordingly, in the AMBITION clinical trial, the combination of an ERA and a PDE5i showed clinical benefits compared to both drugs in monotherapy [56].

### 3.6. Limitations

One limitation of this review is the absence of statistical tests. Many studies did not present the absolute values, and considering the high heterogeneity in the methods, statistic comparisons would be hard to interpret. Also, this study does not include all drugs tested on PAB models, and some research groups may have published the effects of a given drug on different models in different papers. Although it would be interesting to have a picture of all potentially cardioprotective drugs, analysing only studies which test two or more models allows us to understand, for each molecule, which seem to have direct, indirect, or mixed cardioprotective effects. Furthermore, as the same research group performs the experiments on both models, heterogeneity is lower. One more limitation is the absence of drugs which are known to lack benefits on PAH. They would be useful for comparison purposes. In this study, only dapagliflozin completely lacked benefits, but even this drug showed improvements in animal models and in some patients. Finally, this review does not include all the results of the studies: some important outcomes may have been missed and the proportion of similarities/differences between models can be unrepresentative of the full results.

## 4. Materials and Methods

We conducted a comprehensive literature search using online databases, including Scopus, Web of Science, and PubMed/MEDLINE, without time or language limitation.

The queries used were: (pulmonary hypertension OR pulmonary arterial hypertension) AND (SUGEN OR SU5416 OR (chronic hypoxia) OR monocrotaline OR MCT OR Schistosomiasis OR Schistosoma OR (Endothelin receptor-B) OR ET-B OR Angiopoeitin-1 OR Serotonin OR 5-HTT) AND (PAB OR pulmonary artery banding or PTB or pulmonary trunk banding).

### 4.1. Inclusion and Exclusion Criteria

Inclusion criteria consisted in pharmacological interventions to prevent/reverse PH, tested both on a PAB group and in a PH animal model.

Studies were excluded due to acute interventions (only one treatment administration) and lack of data for PAB group and at least one other PH model.

### 4.2. Data Extraction

All selected studies were carefully reviewed. We extracted data from the most assessed and important outcomes related to RV. Outcomes were divided based on model induction, RV structure, RV systolic function, RV diastolic function, and pulmonary vascular hemodynamics and remodeling in order to facilitate their presentation, although some outcomes are related to more than one domain (e.g., BNP is related to RV structure and diastolic function).

## 5. Conclusions

This review showed that many drugs currently under research for PAH have a cardioprotective effect on animals that may translate to humans, as well as pulmonary vascular hemodynamics and remodeling benefits. However, results frequently differ between studies for the same drugs and some studies show unexpected results when comparing PAB to PH models with similar interventions. This may be partly explained by important differences in the model induction, such as the time between the end of the induction and the start of the treatment, as some studies immediately started treating the animals and others waited up to 4 weeks. In order to improve induction methods for human disease translation, further experimental studies should evaluate single variables, such as PAB needle size, influence of the RV function, and pulmonary hemodynamics—including gene expression and protein analysis. Also, further research should compare the effect of drugs already tested in humans with their effect in different animal models.

## Figures and Tables

**Figure 1 ijms-24-15539-f001:**
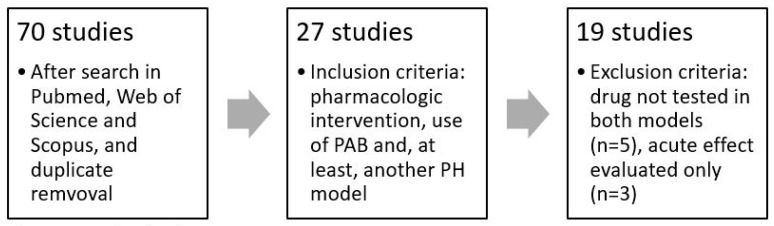
Study selection.

**Table 1 ijms-24-15539-t001:** Selected articles and characteristics.

Study [Reference]	Author	Year	PH Models	Drug(s) Tested	Anaesthetic	Mechanism of Action
1 [13]	Adão, R. et al.	2018	PAB, MCT	Urocortin-2	Sevoflurane	Type 2 CRH receptor activator
2 [14]	Andersen, S. et al.	2019	PAB, SuHx	Sacubitril/valsartan	Sevoflurane	Angiotensin II receptor/neprilysin inhibitors
3 [15]	Budas, G.R. et al.	2018	PAB, MCT, SuHx	GS-444217	Isoflurane (MCT and PAB), xylazine + ketamine (SuHx)	ASK1 inhibitor
4 [16]	Cho, Y.K. et al.	2010	PAB, MCT	Sodium valproate	(no haemodynamic evaluation)	Histone deacetylase inhibitor
5 [17]	Ishii, R. et al.	2020	PAB, MCT, SuHx	Ivabradine	Isoflurane	I_f_ current inhibitor
6 [18]	Kojonazarov, B. et al.	2013	PAB, MCT	Sunitinib, sorafenib	Isoflurane	PDGFR-, VEGFR- and KIT-inhibitor; raf1/b-, VEGFR-, PDGFR-inhibitor
7 [19]	Kurakula, K. et al.	2022	PAB, MCT + shunt	Juglone	ND	Pin1 inhibitor
8 [20]	Kurosawa, R. et al.	2021	PAB, SuHx	Celastrol	Isoflurane	Bsg, CyPA and NF-Kb inhibitor
9 [21]	Li, H. et al.	2021	PAB, MCT	Dapagliflozin	Pentorbital	SGTL-2 inhibitor
10 [22]	Mamazhakypov, A. et al.	2020	PAB, SuHx	Macitentan, tadalafil	Isoflurane	Endothelin-1 receptor antagonist; PDE5 inhibitor
11 [23]	Mendes-Ferreira, P. et al.	2016	PAB, MCT	Neuregulin-1	Sevoflurane, fentanyl, midazolam	ErbB family tyrosine kinase receptors activator
12 [24]	Omura, J. et al.	2020	PAB, MCT	GapmeR H19	Isoflurane	lncRNA H19 suppressor
13 [25]	Pak, O. et al.	2018	PAB, CH	MitoQ	ND	Mitochondria-targeted antioxidant
14 [26]	Piao, L. et al.	2010	PAB, MCT	Dichloroacetate	Isoflurane	Pyruvate dehydrogenase kinase inhibitor
15 [27]	Piao, L. et al.	2012	PAB, MCT	Gallein	Isoflurane	Gβγ–GRK2 signaling inhibitor
16 [28]	Schafer, S. et al.	2009	PAB, MCT	Sildenafil	Pentorbital, isoflurane	PDE5 inhibitor
17 [29]	Sun, X.Q. et al.	2021	PAB, SuHx	Clorgyline	Isoflurane	MAO-A inhibitor
18 [30]	Tanaka, S. et al.	2022	PAB, MCT	Dantrolene	Medetomidin, midazolam, butorphanol	Cardiac ryanodine receptor (RyR2) stabilizer
19 [31]	Van der Feen, D.E. et al.	2019	PAB, MCT + shunt, SuHx	RVX208	ND	BET inhibitor (BRD4 antagonist)

Anaesthetic: anaesthetic used for hemodynamical evaluation; CH: chronic hypoxia; MCT: monocrotaline; ND: no data; PAB: pulmonary artery banding; PH: pulmonary hypertension; SuHx: Sugen 5416/hypoxia.

**Table 2 ijms-24-15539-t002:** Methods overview of studies using MCT or MCT + shunt models.

Study	Intervention (Drug)	Model MCT	Model PAB	Sex MCT	Sex PAB	MCT Dose (mg/Kg)	PAB Needle/Clip Size	Induction to Intervention Period MCT	Induction to Intervention Period PAB	Intervention Period MCT	Intervention Period PAB
1 [13]	Urocortin-2	WR	=	Male	=	60	16 G	2 weeks	=	10 days	=
3 [15]	GS-444217	SDR	CM	Male	=	60	Clip 0.3 mm	1 week	=	3 weeks	2 weeks
4 [16]	Sodium valproate	SDR	=	Male	=	30	22 G	0 days	0 days	3 weeks	=
5 [17]	Ivabradine	SDR	=	Both	=	60	18 G	2 weeks	=	3 weeks	=
6 [18]	Sunitinib/Sorafenib	SDR	=	Both	=	60	Clip 0.9 mm	3 weeks	2 weeks	2 weeks	=
7 [19]	Juglone (shunt)	WR	=	Male	=	60	18 G	3 weeks	4 weeks	2 weeks	4 weeks
9 [21]	Dapagliflozin	SDR	=	Male	=	60	18 G	2 weeks	=	3 weeks	2 weeks
11 [23]	Neuregulin-1	WR	=	Male	=	60	16 G	2 weeks	=	1 week	=
12 [24]	GapmeR H19	SDR	=	Male	=	60	19 G	2 weeks	3 weeks	2 weeks	5 weeks
14 [26]	Dichloroacetate	SDR	=	Male	=	60	16 G	10 days	0 days	3 weeks	7 weeks
15 [27]	Gallein	SDR	=	Both	=	60	18 G	2 weeks	=	2 weeks	=
16 [28]	Sildenafil	SDR	WR	Male	=	60	18 G	2 weeks	0 days	2 weeks	3 weeks
18 [30]	Dantrolene	SDR	=	Male	=	60	18 G	0 days	−1 week	4 weeks	=
19 [31]	RVX208 (shunt)	WR	=	Male	=	60	ND	3 weeks	4 weeks	2 weeks	4 weeks

=: same as MCT values; CM: C57BL/6 mice; MCT: monocrotaline; ND: no data; PAB: pulmonary artery banding; SDR: Sprague/Dawley rats; WR: wistar rats. Induction to intervention period MCT: time from monocrotaline injection until the start of the treatment. Induction to intervention period PAB: time from PAB surgery to the start of the treatment (in one study, treatment started before surgery, so this value is negative).

**Table 3 ijms-24-15539-t003:** Methods overview of studies using CH or SuHx models.

Study	Intervention (Model)	Model CH/SuHx	Model PAB	Sex CH/SuHx	Sex PAB	SU5416 Dose (mg/Kg)	Hipoxia Time (Weeks)	PAB Needle/Clip Size	Induction to Intervention Period CH/SuHx	Induction to Intervention Period PAB	Intervention Period CH/SuHx	Intervention Period PAB
2 [14]	Sacubitril/valsartan (SuHx)	SDR	WR	Male	=	25	4	Clip 0.7 mm	2 weeks	=	5 weeks	=
3 [15]	GS-444217 (SuHx)	SDR	CM	Male	=	ND	4	Clip 0.3 mm	−4 weeks	1 week	4 weeks	2 weeks
5 [17]	Ivabradine (SuHx)	SDR	=	Both	=	20	3	18 G	0 weeks	2 weeks	3 weeks	=
8 [20]	Celastrol (CH/SuHx)	SDR	CM	Male	=	-/20	4/3	25 G	−4/0 weeks	0 weeks	4/2 weeks	3 weeks
10 [22]	Macitentan/Tadalafil/Macitentan + Tadalafil (SuHx)	WKR	=	Male	=	20	3	18 G	2 weeks	1 week	2 weeks	=
13 [25]	MitoQ (CH)	CM	=	Both	=	-	4	Clip 0.35 mm	−4 weeks	0 weeks	4 weeks	=
17 [29]	Clorgyline (SuHx)	SDR	WR	Male	=	25	4	Clip 0.6 mm	4 weeks	2 weeks	3 weeks	6 weeks
19 [31]	RVX208 (SuHx)	SDR	WR	Male	=	20	3	ND	3 weeks	4 weeks	4 weeks	=

=: same as CH/SuHx values; CH: chronic hypoxia; CM: C57BL/6 mice; ND: no data; PAB: pulmonary artery banding; SDR: Sprague/Dawley rats; SuHx: Sugen 5416/hypoxia; WKR: wistar/kyoto rats; WR: wistar rats. Induction to invervention period CH/SuHx: time from the end of the hypoxia period to the start of the treatment (some studies start the intervention during the hypoxia period; in such cases this value is negative). Induction to invervention period PAB: time from PAB surgery to the start of the treatment.

**Table 4 ijms-24-15539-t004:** Results of studies related to RV structure.

		Fulton Index				RV Fibrosis				CSA/D				BNP/NT-proBNP			
Study	Drug(s)	C	M	S	P	C	M	S	P	C	M	S	P	C	M	S	P
1 [13]	Urocortin-2		↓		↓		↓		↓		↓		↓		↓		
2 [14]	Sacubitril/valsartan			↔	↔			↔	↔							↔	↔
3 [15]	GS-444217		↓	↓	↓				↓				↔		↓	↓	
4 [16]	Sodium valproate		↓		↓												
5 [17]	Ivabradine						↓	↓	↓		↓	↓	↓				
6 [18]	Sunitinib		↓		↓		↓		↓				↓		↓		↓
6 [18]	Sorafenib		↓		↓		↓		↓				↓		↓		↓
7 [19]	Juglone		↔		↓				↔								
8 [20]	Celastrol	↓		↓				↓	↓				↓				
9 [21]	Dapagliflozin			↔	↔		↔		↔								
10 [22]	Macitentan			↓	↓			↔	↔							↓ ^#^	↔ ^#^
10 [22]	Tadalafil			↓	↔			↔	↔							↓ ^#^	↔ ^#^
10 [22]	Mac. + Tad.			↓	↔			↔	↔							↓ ^#^	↔ ^#^
11 [23]	Neuregulin-1		↓		↓		↓		↓		↓		↓		↓		
12 [24]	GapmeR H19		↔		↔		↓		↓		↓		↓				
13 [25]	MitoQ	↓			↓												
14 [26]	Dichloroacetate		↓														
15 [27]	Gallein		↔		↔												
16 [28]	Sildenafil		↓		↔						↓		↑		↓		↔
17 [29]	Clorgyline			↓	↔			↔	↔ *			↓	↔				
18 [30]	Dantrolene						↓				↓		↓				
19 [31]	RVX208		↔	↔	↑				↔				↔				

BNP/NT-proBNP: serum brain natriuretic peptide/N-terminal brain natriuretic peptide; CSA/D: cardiomyocyte cross-sectional area/diameter; C: chronic hypoxia; M: monocrotaline (with or without shunt); S: Sugen 5416/hypoxia; P: pulmonary artery banding. ↓—significant decrease in the parameter; ↑—significant increase in the parameter; ↔—no significant effect in the parameter; *—in the animal model, the parameter did not significantly change; ^#^—NT-proBNP.

**Table 5 ijms-24-15539-t005:** Results of studies related to RV systolic function and MAP.

		RVSP				TAPSE				CO				CI				MAP			
Study	Drug(s)	C	M	S	P	C	M	S	P	C	M	S	P	C	M	S	P	C	M	S	P
1 [13]	Urocortin-2		↓		↔		↑				↑		↔ *								
2 [14]	Sacubitril/valsartan			↓	↔			↔	↔							↔	↔			↓	↓
3 [15]	GS-444217				↔				↑		↑		↑						↔	↔	↔
4 [16]	Sodium valproate																				
5 [17]	Ivabradine		↔	↔	↔		↔	↑	↑		↑	↑	↑								
6 [18]	Sunitinib		↓		↔		↑		↑						↑		↑		↔		↔
6 [18]	Sorafenib		↓		↔		↑		↑						↑		↔		↑		↔
7 [19]	Juglone								↔								↔				
8 [20]	Celastrol	↓		↓	↔			↑	↑			↑	↑								
9 [21]	Dapaglifozine		↔		↔		↔		↔												
10 [22]	Macitentan			↓	↔			↑	↑			↑	↔							↔	↔ *
10 [22]	Tadalafil			↓	↔			↑	↔			↑	↔							↔	↔ *
10 [22]	Mac. + Tad.			↓	↔			↑	↔			↑	↔							↔	↔ *
11 [23]	Neuregulin-1		↓								↑		↔								
12 [24]	GapmeR H19		↔		↔		↑		↑		↑		↑								
13 [25]	MitoQ	↔			↔	↔			↑	↔			↔								
14 [26]	Dichloroacetate		↓		↓						↑		↑								
15 [27]	Gallein		↔		↔		↑				↑		↔		↔		↑				
16 [28]	Sildenafil		↓		↔										↑		↔		↔		↔
17 [29]	Clorgyline			↓	↔			↔	↔			↔									
18 [30]	Dantrolene		↓		↔						↑										
19 [31]	RVX208			↓					↔			↔	↔								

CI: cardiac index; CO: cardiac output; MAP: mean arterial pressure RVSP: right-ventricular systolic pressure; TAPSE: tricuspid annular plane systolic excursion; C: chronic hypoxia; M: monocrotaline (with or without shunt); S: Sugen 5416/hypoxia; P: pulmonary artery banding. ↓—significant decrease in the parameter; ↑—significant increase in the parameter; ↔—no significant effect in the parameter; *—in the animal model, the parameter did not significantly change.

**Table 6 ijms-24-15539-t006:** Results of studies related to RV diastolic function.

		RVEDD				RVEDP				Tau			
Study	Drug(s)	C	M	S	P	C	M	S	P	C	M	S	P
1 [13]	Urocortin-2		↓		↔		↓		↔ *		↓		
2 [14]	Sacubitril/valsartan							↔	↔				
3 [15]	GS-444217				↓								
4 [16]	Sodium valproate		↔ *		↓								
5 [17]	Ivabradine		↔	↓	↔		↔ *	↓	↓		↓	↓	↓
6 [18]	Sunitinib		↓		↓								
6 [18]	Sorafenib		↓		↓								
7 [19]	Juglone												
8 [20]	Celastrol			↓					↓				
9 [21]	Dapagliflozin												
10 [22]	Macitentan			↓	↓								
10 [22]	Tadalafil			↓	↔								
10 [22]	Mac. + Tad.			↓	↔								
11 [23]	Neuregulin-1		↓				↓				↓		
12 [24]	GapmeR H19		↓		↓		↓		↓				
13 [25]	MitoQ	↓			↓								
14 [26]	Dichloroacetate												
15 [27]	Gallein												
16 [28]	Sildenafil						↓		↔				
17 [29]	Clorgyline			↔	↔								
18 [30]	Dantrolene						↔		↓		↓		↓
19 [31]	RVX208												

RVEDD: right-ventricular end-diastolic diameter; RVEDP: right-ventricular end-diastolic pressure; Tau: right-ventricular relaxation time constant; C: chronic hypoxia; M: monocrotaline (with or without shunt); S: Sugen 5416/hypoxia; P: pulmonary artery banding. ↓—significant decrease in the parameter; ↔—no significant effect in the parameter; *—in the animal model, the parameter did not significantly change.

**Table 7 ijms-24-15539-t007:** Results of studies related to pulmonary vascular hemodynamics and remodelling.

		PVR			PAAT			Complete Muscularization			Medial/Wall Thickness			TPR		
Study	Drug(s)	C	M	S	C	M	S	C	M	S	C	M	S	C	M	S
1 [13]	Urocortin-2		↓									↓ ^1^				
2 [14]	Sacubitril/valsartan												↓ ^2^			
3 [15]	GS-444217		↓							↓						
4 [16]	Sodium valproate															
5 [17]	Ivabradine															
6 [18]	Sunitinib					↑			↓						↓	
6 [18]	Sorafenib					↑			↓						↓	
7 [19]	Juglone											↔ ^1^				
8 [20]	Celastrol						↑						↓ ^3^			
9 [21]	Dapagliflozin					↔						↔ ^4^				
10 [22]	Macitentan									↔						↓
10 [22]	Tadalafil									↔						↓
10 [22]	Mac. + Tad.									↓						↓
11 [23]	Neuregulin-1		↓			↑						↓ ^4^				
12 [24]	GapmeR H19											↔ ^5^			↔	
13 [25]	MitoQ							↔								
14 [26]	Dichloroacetate					↑										
15 [27]	Gallein															
16 [28]	Sildenafil															
17 [29]	Clorgyline												↔ ^6^			↓
18 [30]	Dantrolene											↓ ^7^				
19 [31]	RVX208		↔	↓								↔ ^1^	↔ ^1^			

PAAT: pulmonary artery acceleration time; PVR: pulmonary vascular resistance; TPR: total pulmonary resistance; C: chronic hypoxia; M: monocrotaline (with or without shunt); S: Sugen 5416/hypoxia; ^1^ medial thickness of small vessels (<50 µm); ^2^ reduction in wall thickness in arterioles of 30–60 µm, but not <30 µm and >60 µm; ^3^ medial thickness of distal pulmonary arteries (50–100 µm); ^4^ pulmonary arterial medial wall thickness; ^5^ medial thickness of small vessels (<100 µm); ^6^ clorgyline reduced intimal thickness, but not medial thickness in pulmonary arterioles (25–100 µm); ^7^ medial wall thickness. ↓—significant decrease in the parameter; ↑—significant increase in the parameter; ↔—no significant effect in the parameter.

## Data Availability

Data is contained within the article or Appendix A.

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
