# Peer review of "Therapeutic Approaches in Pulmonary Arterial Hypertension with Beneficial Effects on Right Ventricular Function—Preclinical Studies"

_ijms, 2023, doi:10.3390/ijms242115539_

Round 1

Reviewer 1 Report

The manuscripts reviewed 19 published studies in which the effect of multiple drugs on PH animal models (chronic hypoxia, or MCT, or Sugen and hypoxia) versus pulmonary artery banding (PAB) models were reported. The authors did a good job in summarizing the similar and different effects of the drug on PH models versus on PAB model, which is an important PH research area. The writing in general is excellent too. I have a few suggestions/comments that may improve the readability of this review.

1.     The Tables are organized as articles (number 1 to 19). However, in the paper (text), the cited paper (for example, Article 1=paper 11) was referred. Please add the referred paper number in the Tables too, so the readers will have less difficulty to follow.

2.     The description of the Tables, particularly for Tables 5-6, should focus more on the differences between the effect on PH models versus on PAB model.

3.     The paragraph at lines 142 to 150, should move to the end as a summary, and replace it with an instruction.

I have no concerns on the quality of English language.

Reviewer 2 Report

Main Question Addressed by the Research:

The main question addressed by the research is relevant, particularly to understanding potential treatments for pulmonary hypertension, as RV function is a key determinant of prognosis in this condition.

Originality:

The topic appears to be moderately original, as it focuses on evaluating the effects of various drugs in different animal models to understand their cardioprotective potential. The originality arises from the comparison of results between different models, which helps identify potential targets for treatment.

I have some suggestions to consider:

-          The manuscript needs to be rearranged; the Materials and Methods section should be placed after the Introduction, following the general presentation guidelines of the journal.

-          In Table 1, I think it would be appropriate to introduce a column that correlates the articles with the citations in the bibliography.

Consistency of Conclusions:

-          The research appears to discuss variations in drug effects across different animal models and parameters, which suggests a degree of inconsistency.  However, it does not specifically outline specific future research directions or suggest potential areas for further investigation.

Minor editing of English language required. 
